# Direction-dependent arm kinematics reveal optimal integration of gravity cues

Jeremie Gaveau[1]\*, Bastien Berret[2,3], Dora E Angelaki[4†], Charalambos Papaxanthis[1†]

[1]Université Bourgogne Franche-Comté, INSERM CAPS UMR 1093, Dijon, France; [2]CIAMS, Université Paris-Sud, Université Paris Saclay, Orsay, France; [3]CIAMS, Université d'Orléans, Orléans, France; [4]Department of Neuroscience, Baylor College of Medicine, Houston, United States

**Abstract** The brain has evolved an internal model of gravity to cope with life in the Earth's gravitational environment. How this internal model benefits the implementation of skilled movement has remained unsolved. One prevailing theory has assumed that this internal model is used to compensate for gravity's mechanical effects on the body, such as to maintain invariant motor trajectories. Alternatively, gravity force could be used purposely and efficiently for the planning and execution of voluntary movements, thereby resulting in direction-depending kinematics. Here we experimentally interrogate these two hypotheses by measuring arm kinematics while varying movement direction in normal and zero-G gravity conditions. By comparing experimental results with model predictions, we show that the brain uses the internal model to implement control policies that take advantage of gravity to minimize movement effort.

*For correspondence: jeremie. gaveau@u-bourgogne.fr

†These authors contributed equally to this work

Competing interests: The authors declare that no competing interests exist.

## Introduction

It is always fascinating to witness the ability of acrobats and dancers to accomplish complex and elegant movements, graciously interacting with gravito-inertial forces. Computational theory postulates that this captivating performance is due to the ability of the brain to learn and store internal representations of environmental dynamics (*Wolpert and Ghahramani, 2000*). On earth, gravity is the most ubiquitous and constant environmental feature. As such, a neural representation of gravity is created and stored through an internal model (*Papaxanthis et al., 1998a*; *Angelaki et al., 1999*; *Merfeld et al., 1999*; *McIntyre et al., 2001*; *Angelaki et al., 2004*; *Indovina et al., 2005*; *Miller et al., 2008*; *Crevecoeur et al., 2009*; *Gaveau and Papaxanthis, 2011*; *Laurens et al., 2013a, 2013b*).

The need for an internal model of gravity arises because Einstein's equivalence principle prevents any single sensory receptor from encoding gravity without simultaneously also encoding inertial accelerations (*Einstein, 1908*). The neural representation of gravity is thought to solve this ambiguity by multisensory statistical inference (*Angelaki et al., 1999*; *Merfeld et al., 1999*; *Angelaki et al., 2004*; *Laurens et al., 2013b*). An internal model of gravity has been shown to benefit the anticipation of a free falling object motion (*Zago and Lacquaniti, 2005*; *Zago et al., 2008*; *Lacquaniti et al., 2013*), as well as the visual perception of allocentric vertical (*Van Pelt et al., 2005*; *De Vrijer et al., 2008*; *Elmore et al., 2014*). However, whether and how an internal model of gravity benefits the planning and execution of skilled movement remains unknown.

One influential viewpoint assumes a need for *compensation*; i.e., the internal model of gravity is used to predict and compensate for its mechanical effects on the body. This hypothesis has been motivated by experimental findings on arm movements in the presence of externally applied Coriolis, viscous force fields and interaction torques (*Shadmehr and Mussa-Ivaldi, 1994*; *Gribble and*

**eLife digest** Many of the activities of humans and other animals require the limbs to be moved in a coordinated manner. For a movement to be successful, the brain must generate muscle contractions that take into account factors in the environment that might affect the movement. One such prominent environmental feature is gravity, and it is broadly believed that the brain develops and uses an internal representation of gravity to anticipate its effects on the limbs.

How an internal representation of gravity helps limb movements to be made successfully is not known. Theorists have proposed that the brain could use the internal model of gravity to predict how to compensate for its mechanical effects – or, on the contrary, take advantage of them.

Flying a plane in a "parabolic" arc creates a microgravity environment inside it that produces a feeling of weightlessness. Gaveau et al. asked volunteers to perform arm movements in normal earth gravity and in microgravity conditions. Under normal gravity, the volunteers made arm movements with speed profiles that differed according to movement direction. When they first performed these movements in microgravity, the speeds still differed according to direction. However, as the participants gained more experience of making the movements in microgravity, the speed at which upward and downward arm movements were made became more similar. Eventually movements were performed at the same speed in either direction.

Comparing these results to numerical simulations revealed a sophisticated behavior where movements are organized to take advantage of the effects of gravity to minimize the effort that the muscles need to make. Further research into the neural mechanisms behind this optimization process could benefit the development of various rehabilitative and assistive technologies, such as brain-machine interfaces and robotic devices to guide and support limbs.

*Ostry, 1999*; *Pigeon et al., 2003*). The main benefit of a compensation strategy would be the simplification of motor planning by allowing for invariant trajectories (*Hollerbach and Flash, 1982*; *Atkeson and Hollerbach, 1985*). Current research in fields as diverse as neurorehabilitation, movement perception, or motor control modularity, assumes such a compensation principle (*Prange et al., 2009*, *2012*; *Cook et al., 2013*; *Russo et al., 2014*).

An alternative perspective is based on a need for *effort optimization,* i.e., the internal model of gravity could be used to predict and take advantage of its mechanical effects on the body. It has been proposed that adaptation to the Earth's gravity field has allowed control policies to evolve that take advantage of environmental dynamics – use gravity as an assistive force to accelerate downward movements and as a resistive force to decelerate upward movements. This strategy, which has been formalized into a *Minimum Smooth-Effort* model, would result in movement kinematics that varies with direction (*Berret et al., 2008a*; *Gaveau et al., 2014*). Indeed, upward movements were shown to have shorter time to peak velocity and larger curvature than downward movements, but such comparisons were until now largely qualitative (*Papaxanthis et al., 1998b*; *Gentili et al., 2007*). Furthermore, upward/downward direction-dependent kinematics could also arise from the complex dynamics of the peripheral neuromuscular system. For example, the firing properties of extensor motoneurons (pulling the arm downwards when upright) are known to obey different rules from those of flexor motoneurons (pulling upwards when uptight; *Cotel et al., 2009*; *Wilson et al., 2015*). In addition, muscle force production generated by eccentric contraction (elongation, e.g. downward movements for flexors) is also known to obey differing rules from concentric contraction (e.g. upward movement for flexors; for a review see *Enoka, 1996*). Because of these asymmetrical neuromuscular peripheral properties, upward/downward kinematic asymmetries cannot necessarily be attributed to gravity effort optimality (i.e., minimization of muscular force to elevate and lower the arm). Thus, a solid test of the effort optimization hypothesis has been lacking.

Here, we explicitly distinguish between the compensation and effort optimization hypotheses with two critical experiments. First, we contrast predictions of optimal control models that either compensate or take advantage of gravity force effects. Then, we quantitatively compare these predictions with actual kinematic features of arm movements in multiple directions. Second, in order to discard a possible influence of peripheral neuromuscular mechanisms, we measure how differences

in upward versus downward arm movement kinematics are influenced by the lack of gravity during the zero-G phase of parabolic flight. If directional asymmetries originate from asymmetric firing of flexor/extensor motoneurons, they should persist in the absence of gravity because motoneurons properties are not expected to change during the short zero-G phase of a parabolic flight (*Ishihara et al., 1996*, *2002*; for a review see *Nagatomo et al., 2014*). On the other hand, if they originate from eccentric versus concentric force production differences, directional asymmetries should cease to exist instantly in 0g because movement dynamics no longer differ for upward and downward movements (*Enoka, 1996*). Alternatively, however, if directional asymmetries originate from neural planning processes that take advantage of the internal model of gravity, following a gradual recalibration of the gravity internal model, directional asymmetries should progressively decrease towards new optimal zero-G values (*McIntyre et al., 2001*; *Izawa et al., 2008*; *Snaterse et al., 2011*). In support of the effort optimization hypothesis, we show that directional asymmetries are gradually eliminated during repeated exposure to the zero-G phase of parabolic flight.

## Results

We asked fifteen humans to perform arm movements around the shoulder joint in different directions (*Figure 1A*), whereby the work of gravity torque is systematically varied (*Figure 1B*), while other dynamic variables (interaction torques, Coriolis and centripetal forces) are constant (see *Figure 1—figure supplement 1* for a geometrical illustration of the mechanical system and details on gravity torque computations). In single-degree of freedom movements, although the spatial shape of the endpoint trajectory is circular and constant across directions, the temporal shape of the endpoint trajectory (the form of the velocity profile) could change. As a measure of the shape of the velocity profile, we define a symmetry ratio (SR: acceleration time divided by the total movement time). SR corresponds to the relative timing of peak velocity (as in *Figure 1C*) and thus allows quantifying whether arm kinematics remains invariant (*compensation* hypothesis prediction) or changes (*effort optimization* hypothesis prediction) with movement direction. Here we use the optimal control framework (*Bryson and Ho, 1975*; *Todorov, 2004*) to simulate arm movement planning in different directions. Specifically, we change the cost function being minimized to understand how the neural model of gravity serves motor planning; i.e., we compare simulations of optimal control models that use the internal model of gravity to either compensate or take advantage of its mechanical effects on the body.

Minimizing a kinematic cost only, without taking joint torque into account, such as the *Jerk model* (*Flash and Hogan, 1985*), is a perfect example of the compensation strategy. This is because the brain must compensate all perturbing dynamics to produce a consistent and invariant kinematically-defined motor plan (*Hollerbach and Flash, 1982*; *Atkeson and Hollerbach, 1985*), thus predicting constant SR values for all movement directions (*Jerk* prediction in *Figure 1D*). In contrast, the *Smooth-Effort* model minimizes a hybrid cost that, in addition to the *Jerk*, takes the external dynamics into account to minimize the muscular force needed to move the arm (absolute work of muscular torque). Thus, by design, the *Smooth-Effort* model implements the effort optimization strategy, whereby the brain uses the gravity internal model to predict and take advantage of gravity torques in accelerating and decelerating downward and upward movements, respectively (*Berret et al., 2008a*, *2008b*; *Gaveau et al., 2011*, *2014*). The effort optimization hypothesis predicts a sigmoidal dependence of SR on movement direction (*Smooth-Effort* prediction in *Figure 1D*). Critically, increasing the weight of movement smoothness (*Jerk*) in the *Smooth-Effort* hybrid cost, leads to a progressive disappearance of the gravity-torque related tuning of arm kinematics (*Figure 1—figure supplement 2*). Accordingly, the compensation hypothesis predicts that movement kinematics remains unchanged for various gravity torque conditions (*Jerk* prediction in *Figure 1E*), whilst the effort optimization hypothesis predicts that movement kinematics strongly correlates (average R = 0.99 [min: 0.986, max: 0.994], p<1e-07) with the amount of gravity torque engaged in the motion (*Smooth-Effort* prediction in *Figure 1E*). Importantly, the SR dependence on movement direction (i.e., on gravity torque) is a unique feature of effort-related optimization as minimizing other cost functions that take joint torques into account, but are not directly related to effort, such as the *Variance* (*Harris and Wolpert, 1998*) or *Torque Change* (*Uno et al., 1989*), predict constant SR as a function of arm movement direction (*Figure 1F*; see also *Figure 1—figure supplement 3* for

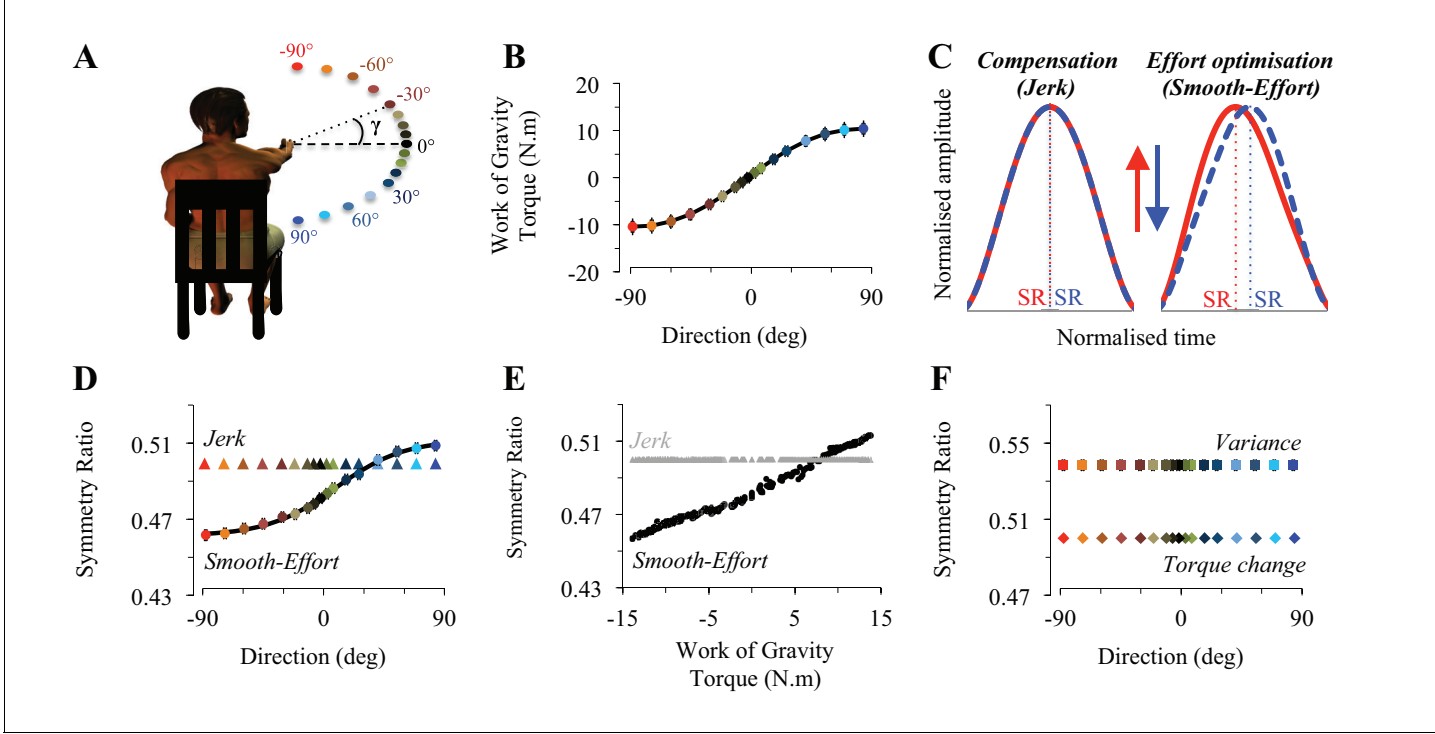

**Figure 1.** Task design and theoretical prediction. (**A**) Participants' initial position and projection of the 17 targets onto the frontal plane. The angle γ, representing the target inclination with respect to horizontal, was used to calculate the gravity torque projecting onto the plane of motion (see also *Figure 1—figure supplement 1*). (**B**) The Work of Gravity Torque (WGT), projected onto the plane of motion and integrated over the whole movement, is plotted as a function of movement direction (same color code as in **A**). This non-linear (cosine-tuned) dependence is well fitted by a sigmoidal function (black curve, average RMSE = 0.12; [min: 0.09, max: 0.15], see Materials and methods). Positive/negative values indicate that WGT has the same/opposite direction as the arm movement. (**C**) Mean velocity profiles (for the average subject), normalized in both amplitude and duration, illustrate the predictions of the compensation hypothesis (*Jerk*) and the effort optimization hypothesis (*Smooth-Effort*) in the vertical plane (upwards, −90; downwards, 90°; see arrows and color-coded direction definition in panel **A**). SR up (red) and SR down (blue) illustrate the calculation of a symmetry ratio (acceleration time / movement time) allowing quantification of kinematic differences/similarities. (**D**) Simulated symmetry ratio predicted by the compensation hypothesis: *Minimum Jerk* (triangles, *Flash and Hogan, 1985*); and the effort optimization hypothesis: *Minimum Smooth-Effort* model (dots, *Gaveau et al., 2014*); as a function of movement direction (−90°, upwards and 90°, downwards). Similarly to WGT (see panels **B**), simulated symmetry ratio obtained from the *Smooth-Effort* model is well fitted by a sigmoidal function (black curve). (**E**) Simulated symmetry ratio as a function of WGT (grey triangles, *Jerk* and black dots, *Smooth-Effort*). Each data point represents the prediction for one subject moving in one direction (n = 255 in each plot). It is noticeable that according to the effort optimization hypothesis, arm kinematics (symmetry ratio) should not be invariant but instead linearly correlate with WGT. (**F**) Simulated symmetry ratio predicted by two other well-known models minimizing dynamic cost functions: the *Minimum Variance* (*Harris and Wolpert, 1998*) and the *Minimum Torque Change* (*Uno et al., 1989*). It can be observed that the modulation of kinematics with movement direction is a specific feature of the effort-related optimization only.

The following figure supplements are available for figure 1:

**Figure supplement 1.** Gravity torque projection in Experiment 1.

**Figure supplement 2.** Clarification on the hybrid cost used in the *Smooth-Effort* model.

**Figure supplement 3.** Supplemental minimum Variance simulations including muscle dynamics.

additional simulations testing the effect of including muscle dynamics into the minimization of end-point *Variance*). In fact, minimization of an effort-related cost is a necessary and sufficient condition to predict directional asymmetries in the vertical plane (*Berret et al., 2008a, 2008b*).

Participants accomplished rapid arm movements with single-peaked velocity profiles. Average duration did not vary with movement direction (0.40s ± 0.01, SD; $F_{16,224}=0.67$, p=0.82). Because arm movements were visually guided, systematic and variable errors were small and independent of

movement direction (-2.5°< SE < 2°; VE < 2.5°; $F_{16,224}$=1.13, p=0.32 and $F_{16,224}$=1.02, p=0.44, respectively). As illustrated in *Figure 2A*, SR shows a sigmoidal modulation as a function of movement direction ($F_{16,224}$=28.36, p<0.001). The robustness of this result is illustrated in *Figure 2B*, which shows sigmoidal fits for individual subjects (average RMSE = 9.77e-03 [min: 4.5e-03, max: 1.53e-02]). When SR was regressed against gravity torque, correlation coefficients were high, averaging R = 0.84 [min: 0.67, max: 0.92], p=2.5e-05 (*Figure 2C*). Furthermore, SR was independent of movement duration and movement amplitude (average correlation coefficient for duration R = 0.22 [min: 0.04, max: 0.57], p=0.41, see *Figure 2—figure supplement 1A*; for amplitude R = 0.29 [min: 0.02, max: 0.69], p=0.26, see *Figure 2—figure supplement 1B*). Thus, arm kinematics is selectively modulated according to the gravity torque requirements of the movement, as predicted by the *Smooth-Effort* model and quantified by high correlation coefficients between predicted and experimental SR values (average R = 0.82 [min: 0.61, max: 0.91], p=5.6e-05; compare *Figure 1D and E* to *Figure 2A and C*, respectively). These findings support the effort optimization hypothesis, whereby the brain implements control policies that exploit gravity effects to minimize muscular efforts.

Although undeniably supportive, interpretation of these results is complicated by direction-dependent properties of the peripheral neuromuscular system (*Enoka, 1996*; *Cotel et al., 2009*; *Wilson et al., 2015*). As a further test, exploitation of microgravity environments (e.g., during the zero-G phase of parabolic flight) offers a powerful tool to interrogate neural vs. peripheral origins of the directional asymmetries. This is because during the repeated transitions to zero-G, gravity torque is temporarily eliminated. According to the effort optimization hypothesis, kinematic asymmetries should gradually but systematically decrease to zero, because a progressive re-optimization neural process should take place gradually over multiple zero-G transitions (*McIntyre et al., 2001*; *Izawa et al., 2008*; *Snaterse et al., 2011*). In contrast, a peripheral origin of kinematic asymmetries leads to different predictions. Specifically, if directional asymmetries originate from flexor/extensor motoneuron properties, SR should not change during the zero-G phase of parabolic flight (*Ishihara et al., 1996*, *2002*; *Cotel et al., 2009*; for a review see *Nagatomo et al., 2014*; *Wilson et al., 2015*). Else, if directional asymmetries originate from force production properties, SR asymmetries should be eliminated instantly, not gradually, in 0g (*Enoka, 1996*). A second experiment was designed to test these predictions.

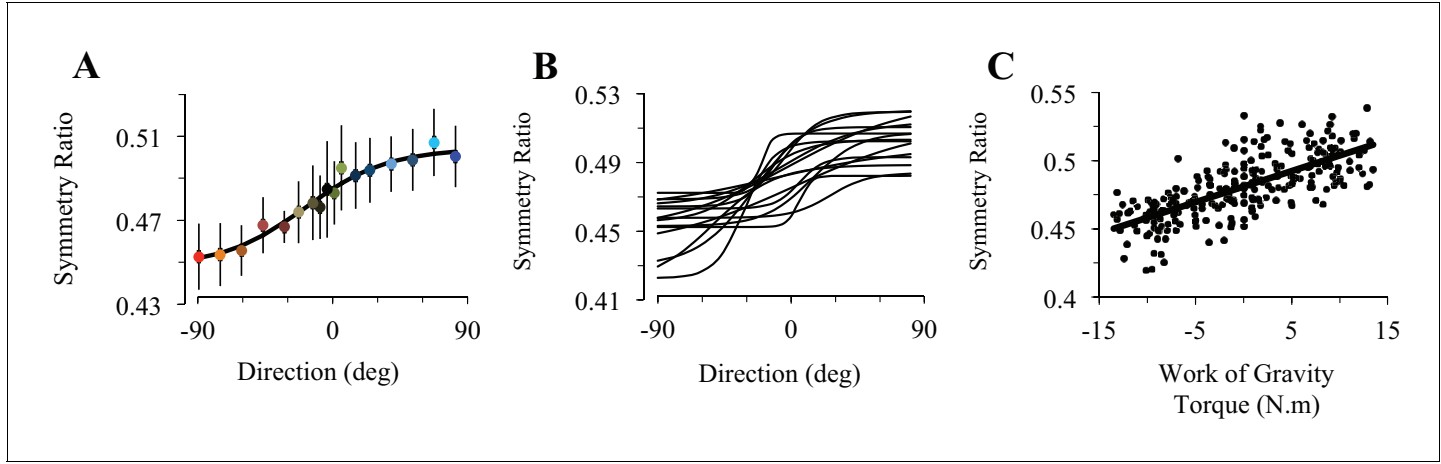

**Figure 2.** Experimental findings. (**A**) Experimentally recorded symmetry ratio, averaged across all subjects, is plotted as a function of movement direction. Similarly to the effort optimization hypothesis prediction (see panels D in *Figure 1*), experimental symmetry ratio is well fitted by a sigmoidal function (black curve). Error bars illustrate SD. (**B**) Fits of a sigmoid function to symmetry ratio as a function of movement direction for data from individual subjects. (**C**) Symmetry ratio as a function of Work of Gravity Torque (WGT). Each data point represents the mean of 12 trials for one subject moving in one direction (n = 255; 3060 trials total). It is noticeable that, similarly to the effort optimization hypothesis prediction (*Figure 1E*), arm kinematics (symmetry ratio) linearly correlates with gravity torque.

The following figure supplement is available for figure 2:

**Figure supplement 1.** Supplemental analyses testing the effect of movement duration and amplitude on experimental findings.

Eleven participants performed fast and visually guided single-degree of freedom arm reaching movements in two directions (toward the head and toward the feet, *Figure 3A*), in zero-G conditions during 5 parabolas (P1-P5) of a flight where centrifugal manoeuvers allow cancellation of gravity effects in the plane's frame of reference. Arm movements were planar with comparable systematic and variable errors for different gravity and direction conditions (shoulder abduction-adduction and internal/external rotation < 3.1°; -3°< SE < 3.3°; VE < 3.4°; gravity effect on SE, p=0.12 and VE, p=0.27; direction effect on SE, p=0.41 and VE, p=0.35). Velocity profiles were single-peaked in both one-G and zero-G conditions and average movement durations ranged between 0.40s and 0.54s (on average, 0.45±0.13s), without any statistical difference between gravity conditions ($F_{5,50}$=1.274, p=0.29) and movement direction ($F_{1,10}$=0.549, p=0.48).

*Figure 3B* shows the SR values predicted from the *Smooth-Effort* model for upward (red) and downward (blue) arm movements, both in one-G and zero-G environments. Optimization to the zero-G environment no longer predicts direction-dependent differences in velocity profiles, as was the case in one-G. In line with the effort optimization hypothesis, which predicts *gradual* adaptation to the zero-G environment, SR slowly converged towards the new direction-independent optimal

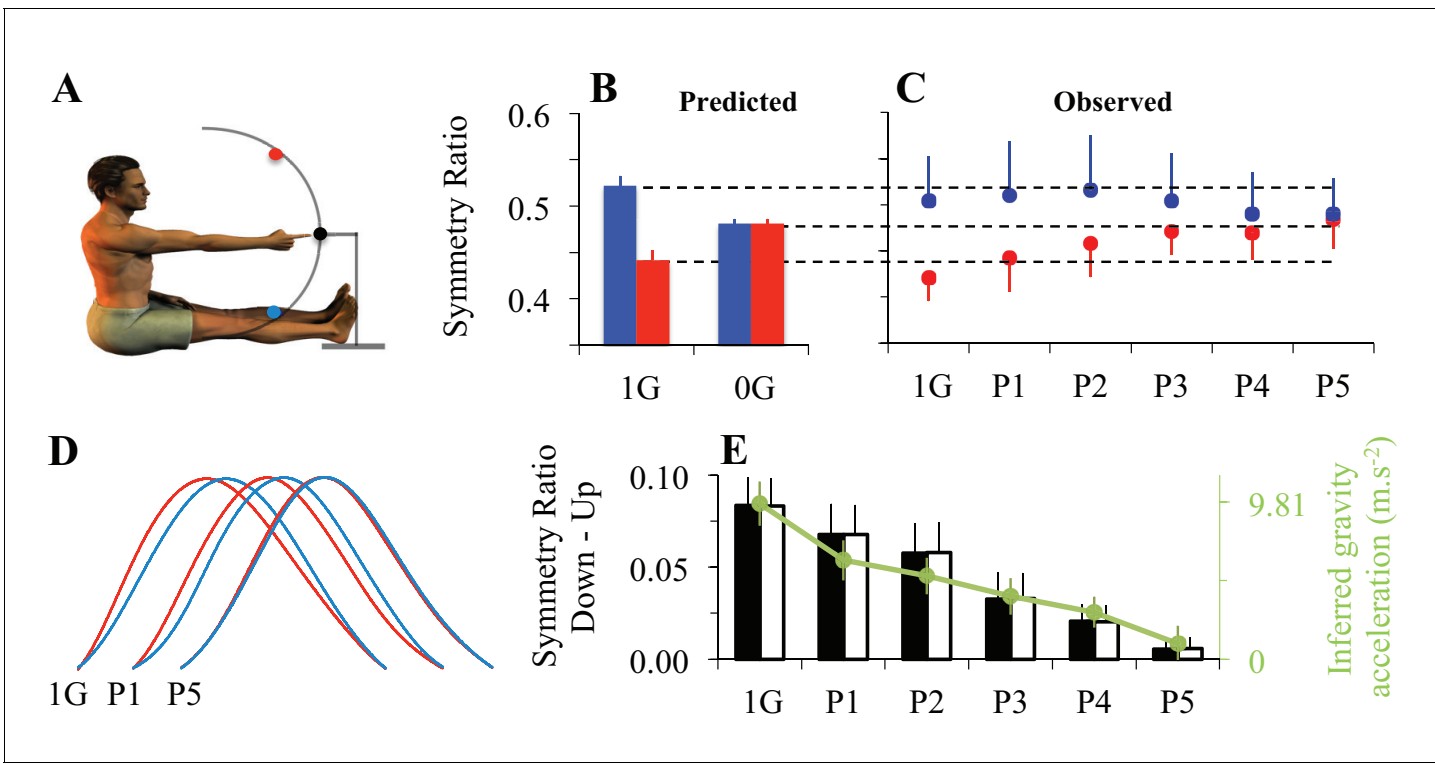

**Figure 3.** Adaptation to microgravity. (A) Participants' initial position and positioning of the 3 targets in the sagittal plane. Eleven participants performed fast and visually guided mono-articular arm movements (shoulder rotations) in the sagittal plane under normal gravity (one-G) and micro-gravity conditions (zero-G) during a parabolic flight (parabola 1, P1 to parabola 5, P5). (B) Symmetry ratios (acceleration time / movement time) predicted by the *Minimum Smooth-Effort* model in one-G and in zero-G conditions. (C) Symmetry ratios experimentally recorded before (1g) and during adaptation to zero-G (P1 to P5). (D) Mean velocity profiles, normalized in amplitude and duration. Qualitative comparisons between upward and downward arm movements illustrate the progressive decrease of directional asymmetries when subjects adapted to the new microgravity environment. (E) Vertical bars represent the average symmetry ratio (black vertical axis, left) for the recorded (black filled bars) and simulated (black open bars) data in one-G and during zero-G (P1 to P5) environments. For simulated data, the g value was fitted (−2*G<g<2*G) in order to best predict the measured symmetry ratios. These fitted g values, represented by the green dots (green vertical axis, right), reveal a progressive decrease of the g internal model value during zero-G exposure. Error bars illustrate SD and color-coded arrows denote movement direction (red = up; blue = down). See also *Figure 3—figure supplement 1*.

The following figure supplement is available for figure 3:

**Figure supplement 1.** Supplemental analysis on the effect of subject order during the microgravity experiment.

value in zero-G (*Figure 3C*; *gravity condition* and *movement direction* interaction effect, $F_{5,50}$=10.54, p<0.001). Importantly, directional asymmetries persisted early in zero-G (p=4.8e-05 for P1 and p=1.09e-03 for P2). This indicates that, during initial exposure to zero-G, the brain still uses the internal model of the one-G environment to plan arm movements.

However, directional asymmetries progressively disappeared in the following parabolas, suggesting a gradual re-optimization of motor commands through sensorimotor adaptation to the zero-G environment (post-hoc, p=0.34 for P3, p=0.92 for P4, and p=0.99 for P5). The corresponding average velocity profiles qualitatively illustrate the increase in acceleration/deceleration symmetry (*Figure 3D*). For the sake of clarity, these experimental findings are replotted as SR Down - SR Up in *Figure 3E* (black bars). To quantitatively validate the hypothesis of a gradual re-optimization of motor commands, we fitted this directional difference with the *Smooth-Effort* model (white bars in *Figure 3E*) by letting the internal model of gravity, g, being a free parameter (-2*G<g<2*G). The progressive decrease in g (internal model of gravity; green dots in *Figure 3E*) shows that the progressive change in arm kinematic asymmetries can be well explained as a recalibration of the gravity internal model used for optimal motor planning based on the effort optimization (minimization) model (*McIntyre et al., 2001*; *Izawa et al., 2008*; *Snaterse et al., 2011*).

## Discussion

It is broadly believed that the brain develops and uses internal models of the sensor and effector dynamics, as well as physical laws of motion, to optimally interact with the external environment (*Shadmehr and Mussa-Ivaldi, 1994*; *Conditt et al., 1997*; *Gribble and Ostry, 1999*; *Wolpert and Ghahramani, 2000*; *Pigeon et al., 2003*; *Todorov, 2004*; *Ahmed et al., 2008*; *Scott, 2012*). Having evolved in the Earth's gravitational environment, our brains have thus acquired an internal model of gravity (*Angelaki et al., 1999*, *2004*; *Indovina et al., 2005*; *Miller 2008*). Here, we have conducted two critical experiments and showed that the brain takes advantage of this internal model to implement control policies that minimize movement effort under gravito-inertial constraints.

First, we have shown that humans use versatile temporal trajectories that are linearly tuned to the gravity torque requirements of the task. Simulations of a *Smooth-Effort* model, which minimizes a hybrid cost composed of the absolute work of muscular forces (mechanical energy expenditure) and jerk (inverse smoothness of the trajectory) can predict not only differences in upward/downward arm movements, but also the linear modulation of endpoint kinematics according to the gravity torque requirements of the movement. Because the smoothness term of the *Smooth-Effort* model corresponds to the *Jerk*, the *Smooth-Effort* model can be considered an expansion of the minimum *Jerk* requirement to include how task dynamics shapes movement kinematics. Yet, inclusion of an effort-related cost is a necessary and sufficient condition to predict directional asymmetries in the vertical plane (*Berret et al., 2008a*, *2008b*). Furthermore, minimizing other torque-related cost functions failed to predict direction-dependent kinematics (*Figure 1F*). The present results therefore strongly support effort minimization in humans, further extending the growing idea that perceived effort plays an important role in the tailoring of human motor as well as non-motor behaviors (*Bramble and Lieberman, 2004*; *Walton et al., 2006*; *Mazzoni et al., 2007*; *Carrier et al., 2011*; *Kurzban et al., 2013*; *Selinger et al., 2015*; *Farshchiansadegh et al., 2016*; *Shadmehr et al., 2016*).

Second, we have also shown that the direction-dependent kinematics observed in normal gravity progressively vanishes during repeated exposure to a microgravity environment. Remarkably, *Smooth-Effort* model simulations nicely predict this adaptation to zero-G. Results of this second experiment are of major importance to disentangle peripheral and neural mechanisms for direction-dependent kinematics. This is because, if the observed direction-dependent kinematics were due to properties of the neuromuscular system, either an abrupt change or no change at all would be expected in the new gravity environment (*Enoka, 1996*; *Ishihara et al., 1996*, *2002*; *Cotel et al., 2009*; *Nagatomo et al., 2014*; *Wilson et al., 2015*). The fact that neither happens reveals a central mechanism.

Furthermore, the fact that the observed progressive changes in kinematic asymmetries lead to new optimal values clearly supports the hypothesis of a progressive re-optimization procedure originating from planning processes (*Izawa et al., 2008*; *Snaterse et al., 2011*; *Selinger et al., 2015*). The present findings on arm movements extend and supplement the recent results of Selinger and

collaborators on human walking (*Selinger et al., 2015*), suggesting that i) the brain can easily and progressively adapt motor patterns to reduce energy expenditure and ii) energy-related criteria (such as effort) are not only the result of, but actually tailor, motor patterns.

The present results and conclusions stand in contrast to a broad view that the brain uses internal models of perturbing forces for their compensation such that stereotypic trajectories can be maintained (*Hollerbach and Flash, 1982*; *Atkeson and Hollerbach, 1985*; *Shadmehr and Mussa-Ivaldi, 1994*). Although very influential, such a compensation hypothesis has been challenged by results of studies that quantified velocity profiles and revealed that the temporal organization of arm kinematics shows a small, yet consistent, dependence on movement direction, speed and load (*Papaxanthis et al., 1998b*; *Gaveau et al., 2011*, *2014*). Furthermore, findings inconsistent with the compensation hypothesis have been largely ignored. For example, *Virji-Babul et al. (1994)* reported regression slopes of SR over movement amplitude that significantly differed between upward and downward movements (see *Figure 2C* in *Virji-Babul et al., 1994*). Also, the consistent observation of negative periods on the phasic activation of arm muscles, resulting from the subtraction of the hypothesized gravity-compensatory activity from the full muscle activation (*Flanders et al., 1996*), suggest that muscular activity does not compensate gravity torque (*Gaveau et al., 2013*).

Luckily, the application of the optimal control theory to the study of biological movement has given better insights into old phenomena. For example, recent studies, which framed motor adaptation as a process of re-optimization (*Izawa et al., 2008*; *Crevecoeur et al., 2009*; *Gaveau et al., 2011*; *Cluff and Scott, 2015*), have reported subtly altered trajectories – by contrast to the traditional compensation view that assumes invariant trajectories. Thus, newly constructed/calibrated internal models may serve trajectory optimization rather than external force compensation. The present results provide further support for this notion and demonstrate the propensity of the motor system for multiple control policies (i.e., trajectories) whose temporal organization shows small, but systematic, differences, such as to allow minimization of motor effort in our daily living ubiquitous gravity environment.

Although we have only used simple, mono-articular arm movements in the present experiments, our conclusions are generalizable. Specifically, directional asymmetries in the vertical plane (upwards versus downwards) have also been observed in multi-articular arm reaching, reaching to grasp, grasping, hand drawing, and whole-body sit-to-stand / stand-to-sit movements (*Papaxanthis et al., 1998c*, *2003*, *2005*; *Yamamoto and Kushiro, 2014*). Thus, we propose that the directional tuning of movement kinematics is a general feature of motor control that may reflect an evolutionary and/ or developmental advantage for effort optimization in the Earth's gravity field (*Bramble and Lieberman, 2004*; *Carrier et al., 2011*; *Selinger et al., 2015*).

Finally, it is important to speculate that the *Smooth-Effort* model pioneered here should not be considered solely an extension of the minimum jerk optimization model, which was proposed for planar horizontal movements that are unaffected by gravity (*Flash and Hogan, 1985*), to now include optimization of work against gravity for vertical movements. Even for movements in the horizontal plane, because the effort component of the cost function in the *Smooth-Effort* model is torque-dependent, the predictions of the Smooth-Effort model will change with the torque requirements of the movement. The minimum *Jerk* model predictions, however, will remain constant. Therefore, if interaction torques (for multi-degree of freedom arm movements) or additional external torques (produced by a robotic manipulandum for example) are experienced in the horizontal plane, the *Smooth-Effort* and *Jerk* model predictions would be different. Future experiments should test whether the need for optimization of work is limited to vertical movements pro or against gravity or, as we propose, represent a more general principle of motor control.

## Materials and methods

Twenty-six right-handed healthy adults participated in these experiments (*Experiment 1*: 4♀ / 11♂, mean age = 24±3.2 years; *Experiment 2*: 2♀ / 9♂, mean age = 27±4.1 years). All gave their written informed consent. Right hand preference was evaluated by the Edinburg test (individual scores > 0.86; *Oldfield, 1971*). The regional ethics committee of Burgundy (C.E.R) approved the protocol of *Experiment 1* and the ethics committee of INSERM (Institut National de la Santé et de la Recherche Médicale) approved the protocol of *Experiment 2*. All procedures were carried out in agreement with local requirements and international norms (Declaration of Helsinki, 1964).

## Experiment 1

Participants comfortably sat on a chair with their trunk in the vertical position (*Figure 1A*). All trials started from a fixed initial position: shoulder elevation 90°, shoulder abduction 0°, elbow joint fully-extended, hand semi-pronated and aligned with the upper arm and the forearm. From that initial position, participants carried out rapid, visually guided, single degree of freedom arm movements (rotation around the shoulder, 45° amplitude) towards 17 targets (plastic markers, diameter 1 cm) placed in the right sagittal-frontal space. Note that results from previous experimental and theoretical studies have demonstrated that directional asymmetries in the vertical plane do not originate from the existence of inertial interaction torques at the elbow and wrist joints (*Le Seac'h and McIntyre, 2007*; *Gaveau et al., 2014*). *Figure 1A* depicts the projection of targets' position onto the frontal plane. The inter-target angles are described in *Figure 1—figure supplement 1C* (Plane angles). All targets were centered on the participants' right shoulder at a distance equal to the length of their fully-extended arm. Reaching movements required a combination of shoulder abduction and shoulder flexion or extension. Participants performed 204 trials in a random order (12 trials for each movement direction, total trials in the experiment = 3060).

## Experiment 2

This experiment took place in an aircraft during parabolic flight. Participants comfortably sat on the aircraft's floor with their legs strapped and their trunk in the vertical position (*Figure 3A*). The general organization of the task was exactly as described in previous studies (*Gaveau et al., 2011*, *2014*). Briefly, 3 targets were centered on the participants' right shoulder (parasagittal plane) at a distance equal to the length of their fully extended arm. Participants were requested to perform fast visually guided upward and downward arm reaching movements (45° shoulder rotation). Participants first performed arm movements in normal gravity during the flight (before parabolic maneuvers started, 40 trials for each movement direction) and then in microgravity during 5 parabolas (≈75 trials, ≈15 trials per parabola). During the parabolic flight and based on pilots' instructions about the gravity force level, the experimenter verbally instructed participants when to start a block of reaching movements and when to stop. This was important to ensure that movements started and finished within zero-G conditions; i.e., participants made no movement in one-G or two-G conditions during or between parabolic manoeuvers. At the beginning of a block of movements within a parabola, participants repetitively reached between the middle, the upward, and the downward targets as follows: middle-upward (stopped for roughly 1 s), upward-middle (pause 1 s), middle-downward (pause 1 s), downward-middle (pause 1 s) and so on for approximately 15 trials. As initial position did not influence adaptation results across parabolas, movements with different initial positions were pooled together within each direction. The experiments were carried out during 4 different flights. Each flight was composed of thirty parabolas, each parabola consisting of three successive phases: (i) hypergravity ∼ 1.8 g, (ii) microgravity ∼ 0 g, and (iii) hypergravity again ∼ 1.8 g. Each of those three phases lasted ∼30 s and the parabolas were separated by a time interval of ∼ 2 min. The whole flight lasted ∼ 2 hr. Here we present the results of an experiment performed during 5 parabolas for each participant. After this experiment, participants carried out a different experiment from which the results are not presented here.

Two participants were tested on each flight. Therefore, to reduce motor adaptation to zero-G before the experiment took place, participants who did not perform the experiment during the first five parabolas were restrained on the aircraft floor so as to prevent any motion. Similar results were observed for participants who did the experiment at the beginning or at the end of the flight (see *Figure 3—figure supplement 1*).

### Data recording, analysis and modeling

All analyses were performed using custom programs in Matlab (Matworks, Natick, MA) and have been described in details in previous studies (*Gaveau et al., 2011*, *2014*). Arm movements in both *Experiments* were recorded using an optoelectronic system of motion analysis (Smart, B.T.S., Italy) with 4 TV-cameras (120 Hz). Five reflective markers (diameter: 4 mm) were placed on the shoulder (acromion), elbow (lateral epicondyle), wrist (middle of the wrist), hand (first metacarpo-phalangeal joint), and the nail of the index. Kinematics was recorded in three dimensions (*X, Y* and *Z*) and low-pass filtered (10 Hz) using a digital fifth-order Butterworth filter. The start and end of each

movement was defined as the time at which finger tangential velocity went above or fell below 5% of maximum velocity. An automatic inspection of all trials revealed that shoulder angular velocity profiles were single-peaked and presented no motion (<3°) from any other joint. Following this analysis, we calculated the subsequent kinematic parameters: (i) movement duration (MD), (ii) constant and variable angular final error and (iii) symmetry ratio (SR) of the finger velocity profile, defined as the ratio of acceleration time to total movement time (a ratio equal to 0.5 indicates temporally symmetric velocity profiles). In the present study, we used SR to quantify kinematic variations with movement direction. SR is a standard parameter that has been routinely used in numerous studies, therefore ensuring an easy comparison of the present result with past ones.

We calculated the gravity torque (GT) normal to the plane of motion with the following formula:

$$GT = mgl\cos\theta\sin\gamma \tag{1}$$

where $m$ is the mass of the arm (estimated for each subject from anthropometric tables, **Winter, 1990**), $g$ = 9.81 m.s$^{-2}$, $l$ is the lever arm length (**Winter, 1990**), $\theta$ is the movement amplitude (0° to 45°) and $\gamma$ is the plane of motion inclination with respect to horizontal. The geometrical illustration of the mechanical system along with details on how to derive **Equation 1** are displayed in **Figure 1— figure supplement 1**.

In **Experiment 1** we report the **Work of Gravity Torque** calculated as follows:

$$WGT = \int_{0}^{45} (GT)\, d\theta \tag{2}$$

Positive values indicate that WGT has the same direction as arm motion, whilst negative values indicate that WGT direction is opposite to arm motion direction (see **Figure 1—figure supplement 1**).

In **Experiment 1,** we also performed iterative least square minimizations to fit a sigmoid function on simulated as well as experimentally recorded SR. This was performed using the "nlinfit" Matlab function (Mathworks) and **Equation 3**:

$$F = p_1 + p_2 \left/ \left( 1 + \exp\left(\frac{-x + p_3}{p_4}\right) \right) \right. \tag{3}$$

where $p_1\ to\ p_4$ are free parameters and $x$ is the angular scale. The goodness of fit was assessed by the Root-Mean-Square Error (RMSE), which is essentially equivalent to Standard Deviation and has the same units as the variable being fitted.

## Simulations

We predicted SR using an optimal control framework based on the *Minimum Smooth-Effort* model (**Gaveau et al., 2014**). More sophisticated versions of the model focusing on multi-degree of freedom arm movements have been described in previous publications (**Berret et al., 2008a**, **2008b**; **Gaveau et al., 2011**).

**Equation 4** describes the equation of motion for a single degree of freedom limb movement, with amplitude angle $(\theta)$, plane of motion inclination with respect to horizontal $(\gamma)$, moment of inertia $(I)$, viscous friction coefficient $B = 0.87$ (as in **Nakano et al., 1999**) and gravitational torque $GT(\theta, \gamma)$. The net muscle torque acting at the shoulder is obtained as follows:

$$\tau = I\ddot{\theta} + B\dot{\theta} + GT(\theta, \gamma) \tag{4}$$

The *Minimum Smooth-Effort* model minimizes a combination of Effort and Smoothness. The mechanical effort related to a movement, i.e., the amount of muscular force spent to move the arm, can be computed as the absolute work of the muscle torque:

$$C_{effort} = \int_{0}^{T} |\tau\dot{\theta}|\, dt \tag{5}$$

The rationale of this effort term is that the desired trajectory must take advantage of non-muscular

torques – gravity torque in the present experiment – in order to minimize the amount of muscular torque required to move the arm up to the final posture.

It has been shown that considering effort expenditure alone usually fails to account for several features of human trajectories, such as motion smoothness (*Berret et al., 2011a*, *2011b*). Consequently, we considered that a complementary objective of motor planning was to maximize motion smoothness. This was achieved by penalizing large angle jerks. Thus an additional term that enters into the minimization is:

$$C_{smooth} = \int_0^T (d\ddot{\theta}/dt)^2 dt \qquad (6)$$

The *Smooth-Effort* model then relies on the following composite cost function:

$$C = C_{effort} + \alpha C_{smooth} \qquad (7)$$

where $\alpha$ is a weighting factor normalizing the relative magnitude of the jerk term in the total cost function. For all simulations, but *Figure 1—figure supplement 2*, we set $\alpha = 7e - 5$. *Figure 1—figure supplement 2* presents simulations where $\alpha$ was systematically varied to test the relative roles of both the *effort* and *smoothness* parts of the cost function in predicting direction dependent kinematics (i.e., the linear correlation of SR with the Work of Gravity Torque; WGT, see *Figure 1E*). Note that minimizing some function $f(x) = b * g(x) + c * h(x)$ will provide the same solution x as minimizing $f(x)/d = b/d * g(x) + c/d * h(x)$ for any d>0. Hence, we normalized our cost function by setting b=d, thereby assuming a unit coefficient in front of one component of the cost.

For a given $\alpha$, the optimal control problem consists of finding a vector (here the time derivative of the muscle torque) driving the system from an initial $(\theta_0)$ to a final static posture $(\theta_f)$, in time $T$ (adjusted for each subject based on experimental data), and yielding a minimum cost value $C$. We solved the minimization numerically using a Gauss pseudospectral method and the software GPOPS (*Benson et al., 2006*; *Garg et al., 2009*; *Rao et al., 2010*). We verified that the control variable was smooth, the boundary values were not reached and the Pontryagin's maximum principle necessary conditions (such as the constancy of the Hamiltonian) met. Predicted SR values were determined based on the simulated velocity profiles, as described above for experimental data.

We also derived the solution of the *Minimum Smooth-Effort* model for a constant gravitational torque and obtained very similar results to those presented in the main text. This means that asymmetries are not totally due to GT variations along the movement amplitude but to the presence of non-zero GT. The linearized case for similar models has been analyzed in depth in *Berret et al. (2008a)*; where the solution of the *Minimum Smooth-Effort* model was derived explicitly and the origin of asymmetries in such a model was mathematically demonstrated – minimizing an effort-related cost is a necessary and sufficient condition for the production of a brief transient muscle inactivation near the peak of velocity which in turn induces the observed kinematic asymmetries with respect to movement direction.

In *Figure 1* we also present the results of simulations performed with three influential alternative models: the *Minimum Jerk*, the *Minimum Torque Change* and the *Minimum Variance* (*Flash and Hogan, 1985*; *Uno et al., 1989*; *Harris and Wolpert, 1998*). Predicted SR values corresponding to the minimization of each subjective cost were obtained using the following equations and the same optimal control framework as described for the *Minimum Smooth-Effort* model.

The cost to minimize for the *Jerk* model was:

$$Cj = \int_0^T (d\ddot{\theta}/dt)^2 dt \qquad (8)$$

As we deal with a 1-dof arm, this model accounts for both the angular and Cartesian versions of the *Minimum Jerk* (which indeed provide equivalent predictions in the current case). Note that the smoothness term of the *Smooth-Effort* model corresponds to the minimum *Jerk*. The *Smooth-Effort* model thus represents an important and non-trivial extension of the minimum *Jerk* model, which also accounts for how task dynamics shapes movement kinematics. Because the effort part of our model is torque dependent, the prediction of the *Smooth-Effort* model will change with the torque

requirement of the movement whilst the minimum Jerk prediction will not. Therefore, if interaction torques (for multi-degree of freedom arm movements) or additional external torques (produced by a robotic manipulandum for example) are experienced in the horizontal plane, the minimum *Smooth-Effort* and the minimum *Jerk* would predict very different solutions.

The cost to minimize for the *Torque Change* model (*Uno et al., 1989*) was:

$$C\tau = \int_0^T \left(\frac{d\tau}{dt}\right)^2 dt \tag{9}$$

Simulations of the Minimum Torque change model for horizontal single degree of freedom arm movements performed under various dynamics by *Tanaka et al. (2004)* reported that the *Minimum Torque Change* model predicts velocity profiles that are always symmetrical, in line with *Figure 1F*. The *Minimum Torque Change* model in the gravity field has also been investigated in depth in (*Berret, 2009*).

We also derived the solutions of the *Minimum Variance* model in the gravity field (*Harris and Wolpert, 1998*). We first linearized the arm's dynamics (*Equation 4*) in order to compute the optimal solution according to the method presented in *Harris and Wolpert (1998)* and *Tanaka et al. (2004)*. Here:

$$mglcos\theta \approx mglcos\theta_0 = k \tag{10}$$

*Equation 10* formalizes that a constant gravitational torque is pushing the moving segment downwards. Note that such linearized case for similar models has been extensively used by previous studies (*Harris and Wolpert, 1998*; *Tanaka et al., 2004*; *Berret et al., 2008a*).

We considered the discrete time version of the *Minimum Variance* optimal control problem. Denoting by $x = (\theta, \dot{\theta}, \tau)^T$ the state vector and by $u = \dot{\tau}$ the control variable, the linear state-space dynamics for the *Minimum Variance* model was expressed as follows:

$$x_{t+1} = Ax_t + B(u_t + w_t) + C \tag{11}$$

where $w_t \sim N(0, \sigma u_t^2)$ is the signal dependent (multiplicative) noise at time $t$ ($\sigma = 0.2$ in all simulations) and $C = (0, -k/I, 0)^T \Delta t$ with $\Delta t$ the time step size after discretization ($\Delta t = 10ms$ in our simulations).

We obtained the solution to the optimal control problem by iteratively computing the distribution of the state vector at time $t$:

$$E[x_t] = A^t x_0 + \sum_{i=0}^{t-1} A^{t-1-i}(Bu_i + C) \tag{12}$$

$$cov[x_t] = \sigma \sum_{i=0}^{t-1} \left(A^{t-1-i}B\right)\left(A^{t-1-i}B\right)^T u_i^2 \tag{13}$$

The positional variance of the endpoint at time t (denoted by $V_t$) is thus given by the element $(1,1)$ of the matrix $cov[x_t]$ (*Equation 13*). We can then define a quadratic programming problem by defining a cost related to the endpoint positional variance as follows:

$$C_{var}(u_0, u_1, \ldots, u_{T+R}) = \sum_{t=T+1}^{T+R} V_t \tag{14}$$

where $R>1$ is an integer defining the post-movement stabilization time, $T$ is the a priori chosen movement time (counting only the transient phase). In our simulations, we considered $R = T$. Because noise accumulates through time, $C_{var}$ is indeed a function of the control variable during the transient period of the motion. This minimization problem was solved using Matlab's *fmincon* function (*sqp* algorithm).

For the sake of completeness, we also derived the minimum variance solution when modeling some basic muscle dynamics, as performed by previous studies (*Harris and Wolpert, 1998*; *Tanaka et al., 2004*). In agreement with previous results from *Tanaka et al. (2004)*; we obtained

similar results to those presented in the main text (*Figure 1F*), as illustrated in *Figure 1—figure supplement 3*. In that case, the state vector becomes $x = (\theta, \dot{\theta}, a_{ag}, a_{ant})^T$, the control variable becomes $u = (u_{ag}, u_{ant})^T$ and the muscles are modeled as first order low-pass filters. The equations for the associated dynamics are as follows:

$$\tau_{ag} - \tau_{ant} = I\ddot{\theta} + B\dot{\theta} + GT(\theta, \gamma) \tag{15}$$

$$\tau_{ag} - \tau_{ant} = \rho(a_{ag} - a_{ant}) \tag{16}$$

$$\sigma\dot{a}_{ag} = u_{ag} - a_{ag} \tag{17}$$

$$\sigma\dot{a}_{ant} = u_{ant} - a_{ant} \tag{18}$$

*Equation 15* describes the equation of motion for a single degree of freedom limb movement, with amplitude angle ($\theta$), plane of motion inclination with respect to horizontal ($\gamma$), moment of inertia ($I$), viscous friction coefficient $B = 0.87$ (*Nakano et al., 1999*) and gravitational torque $GT(\theta, \gamma)$. In *Equation 16*, the constant $\rho$ is a gain factor relating agonist and antagonist muscle activations to joint torques. *Equations 17 and 18* describe the muscle dynamics as a first order low-pass filter. The control variable is the motoneurons inputs $u_{ag}$ and $u_{ant}$ with the constraint: $(u_{ag}, a_{ant}) \in [0, 1]^2$. This implies the positivity of muscle activations and, therefore, muscle torque. The net torque is simply obtained by subtracting the agonist and antagonist torques.

## Statistical analyses

For each participant, we calculated mean values for all recorded variables and checked for normal distribution (Shapiro-Wilk tests) and sphericity (Mauchly tests). In *Experiment 1*, statistical effects were accessed by within-subjects one-way repeated-measures ANOVA (factor: 17 angles). In *Experiment 2*, statistical comparisons were carried out by within-subjects two-way repeated-measures ANOVA. The factors were *direction* (2 levels) and *gravity-conditions* (six levels: 1g and 5 parabolas in 0g). For all statistical analyses, post-hoc differences were assessed with Scheffé's tests and significance was accepted at p<0.05.

## Acknowledgements

This work was supported by the Institut National de la Santé et de la Recherche Médicale (INSERM), by the Agence National de Recherche (ANR, project MOTION ANR-14-CE30-0007-01), by the Centre National d'Etudes Spatiales (CNES) and by National Institute of Neurological Disorders and Stroke Grant R21-NS-075944-02. J Gaveau was supported by grants from the Ministère de l'Éducation Nationale, de l'Enseignement Supérieur et de la Recherche. We would like to thank Yves Ballay, Fabien Nicol and Cyril Sirandré for their help with data acquisition and technical support.

## Additional information

### Funding

| Funder | Grant reference number | Author |
| --- | --- | --- |
| Institut National de la Santé et de la Recherche Médicale | | Jeremie Gaveau Charalambos Papaxanthis |
| Agence Nationale de la Recherche | projet MOTION, 14-CE30-007-01 | Charalambos Papaxanthis |
| National Institute of Neurological Disorders and Stroke | R21-NS-075944-02 | Jeremie Gaveau Dora E Angelaki Charalambos Papaxanthis |
| Centre National d'Etudes Spatiales | | Jeremie Gaveau Bastien Berret Charalambos Papaxanthis |

The funders had no role in study design, data collection and interpretation, or the decision to submit the work for publication.

## Author contributions
JG, Conception and design, Acquisition of data, Analysis and interpretation of data, Drafting or revising the article; BB, Acquisition of data, Analysis and interpretation of data, Drafting or revising the article; DEA, Prepared the manuscript, Drafting or revising the article; CP, Conception and design, Drafting or revising the article

## Author ORCIDs
Jeremie Gaveau, http://orcid.org/0000-0001-8827-1486
Dora E Angelaki, http://orcid.org/0000-0002-9650-8962

## Ethics
Human subjects: Informed consent, and consent to publish, was obtained from all participants. The regional ethics committee of Burgundy (C.E.R) and the ethics committee of INSERM (Institut National de la Santé et de la Recherche Médicale) approved experimental protocols. All procedures were carried out in agreement with local requirements and international norms (Declaration of Helsinki, 1964).

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
