## [Decision Letter]

Thank you for submitting your article "Direction-dependent arm kinematics reveal optimal integration of gravity cues" for consideration by *eLife*. Your article has been reviewed by three peer reviewers, and the evaluation has been overseen by Eve Marder as the Senior Editor. We apologize for the delay in coming to a decision, but it took a while for the reviews to come in, and then the editor and reviewers went through a fairly protracted consultation process to arrive at a decision. Basically, the reviewers all found the topic interesting and important, and felt that your work had a lot to contribute. At the same time, they felt that you have gone too far to claim its novelty and haven't pushed the analyses as far as might be suitable and useful. All of the reviewers agree that a substantial revision is warranted. The comments during the consultation session largely reiterated the comments in the reviews, so I am taking the unusual tack of forwarding the initial reviews, lightly edited, to you, and asking you to do the most thoughtful revision that addresses the substantive issues that the reviewers raised. This will probably require some additional data analysis and rewriting to do more justice to the models and accomplishments already in the literature.

Reviewer #1:

The goal of the present experiment was to settle the long-lasting debate of how the gravitational constraint is integrated in movement control. More specifically, the authors tested whether the motor system attempts to overcome the gravitational force (compensation hypothesis) or rather exploits this passive force to control movement (optimization hypothesis). To disentangle between both hypotheses, the authors analyzed the kinematics of movements performed by human participants in different directions on Earth with predictions made by a mathematical model built on the basis of the optimal control framework. Furthermore, the authors had the unique opportunity to test these hypotheses by asking participants to perform similar movements in absence of gravitational constraint (microgravity), i.e. during parabolic flights. The results of both experiments (normogravity and microgravity) and the comparisons of real and simulated movements in both environments clearly supported the optimization hypothesis.

This study continues the series of nice papers published by the authors on the effects of gravitoinertial constraints on sensorimotor processes. It addresses an important issue related to the integration of gravity during motor actions and provides significant advance over previous studies. However, the paper could be easily improved with some changes.

The problem is that other published papers, some of them referenced in the manuscript, provided convincing evidence in favor of the compensation hypothesis. Unfortunately, the authors do not provide explanations as to why the earlier evidence for the compensation hypothesis is less convincing than theirs. They kind of "kicked the ball into touch" by simply stating that "advent of the optimal control theory has cast strong doubts on this more traditional view of motor control". This was done perhaps on a voluntary basis because critics, even those that lead to progress, are not always welcome. However, because the goal of the study was "to explicitly distinguish between the compensation and optimization hypotheses", the discussion should be more explicit about problems regarding the compensation hypothesis. Despite the very nice results, without such a discussion, the impact of the present paper will be dampened.

The writing style would benefit from smoothing. Currently, there are too many long sentences, many of them are interrupted by long lists of references that sometimes interfere with understanding.

Reviewer #2:

The authors provide interesting evidence of gravitational influence on human single-joint movements. The directional dependence on the ground, and acquired loss of dependence in repeated parabolic flight, is substantive and novel. This work has implication for how the brain can ordinarily (on the ground) and flexibly (via parabolic flight) be influenced by gravity.

The particular analytical, theoretical, and hypotheses offered, however, have several under-developed and contradictory elements that cause great concern. The chief concern, with regard to the overall hypothesis, regards the sureness that these results confirm a "Minimum Smooth Effort" model. The authors choose to cite papers that develop the model in full, rather than doing so here; that is a great disadvantage, as the reader cannot even consider the actual hypothesis suggested. What is suggested throughout is that "effort" is dominated by forces (here, gravity) (e.g. "The behavioral signature of the optimal integration of gravity hypothesis…") The equations offered, however, show that jerk, a purely kinematic measure, is included in the authors' model. The sensitivity analysis, shown only in the discussion, is very incomplete, as it explores the calculated cost only on one side of the chosen coefficient. My best conclusion is that the jerk inclusion is crucial, and even the most generous reading of the work concludes with a synthesis, that both kinematics and dynamics are important in the present behavior.

Several other elements of the analysis, modeling, and interpretation are of concern.

– The presented smooth effort model includes separate terms for torques generated by agonists and antagonists. The authors have no way to measure these; they measure only kinematics; undoubtedly on the ground, and absolutely in flight, there will be co-contraction. The central variables in their model, therefore, are unmeasurable.

– The absolute flat lines calculated from minimum torque change and minimum variance hypotheses (Figure 1) is highly questionable. Similar work (indeed the origin of these theories) suggest that load should play an important role. Much more detail on the calculation of these cost functions is needed to justify this very counter-intuitive result. The minimum variance model, in particular, needs a level of neural activation (drive) that seems complete absent from the present consideration.

– Passengers that are strapped in the flight clearly have cues that gravity is very strong, absent, and strong again. The notion that strapping will lead to no information before movement is vexing. Similarly, the notion that hypergravity for the not-strapped-down has no learning effect is unusual.

– The authors provide scant information on how they calculated just the biomechanics of the task. Meaningful quantitative consideration of kinematics versus dynamics requires more information here.

On the whole, the authors have shown that in a vertical plane, target direction, on the ground, plays a role in human trajectory control; that in early flights, an asymmetry in velocity profiles exist; and in later flights, that asymmetry disappears. All these point to a simple conclusion: the dynamics of gravity can be used by the brain to influence trajectory. All further conclusions made by the manuscript have under-developed evidence, and therefore provide very unclear impact on the field of motor control and neural computation.

Reviewer #3:

General Assessment

This manuscript presents new data and computational analysis supporting the hypothesis that the motor system generates arm movements to satisfy a tradeoff between optimizing smooth kinematics and control effort. The issue of what is being optimized in motor control has been attracting considerable attention through the last decades and no final resolution has been reached to date. Model predictions tend to be consistent with data within specific domains and in some ways this may not be an exception. The authors are considering the impact of gravity in the formation of arm trajectories and they frame the issue in the dichotomy between compensation and exploitation of gravitational torques in the generation of movement. To investigate how the motor system may take advantage of gravity in an optimal control approach, the authors consider a cost function expressing a tradeoff of effort and smoothness terms. They test the model in normal gravity and in microgravity and find evidence for a slow adaptive response when subjects perform repeated movements in the absence of gravity. The key metric considered in this work is the symmetry ratio (SR) expressing the temporal symmetry of acceleration and deceleration phases of simple reaching movements. The authors show that the symmetry ratio (as a function of movement direction and of gravitational work) follows a sigmoidal trend both in the model and in the experiments. Most importantly, the trend is gradually abolished when movements are carried out at zero-g. This is important, in my opinion, because it is not consistent with the possibility that the observed trend in normal gravity is due to an imperfect controller, attempting to implement a kinematic plan with a poor model of the gravitational field. The manuscript is well written, the comparison of the data with the model is done carefully and the conclusion that the data are supporting the idea that the motor controller exploits gravity to shape movement kinematics is convincing.

Substantive concerns

1) Small effect in a limited domain of motor behavior. The effect considered is a relatively subtle asymmetry in the speed profile that is consistent with exploiting the gravitational torque in the upward and downward phase of movement. While the data are consistent with the postulated cost function, the argument about the evolutionary advantage does not seem to match the small size of the described effect. Perhaps the effect on higher dimensional motions would be more substantial.

2) The model does not refute the minimum jerk optimization but it combines it with the optimization of work against gravity. The smoothness model was proposed for planar horizontal movements that are unaffected by gravity. In this respect the model of Gaveau and colleagues does not seem an "alternative" to the minimum jerk model but an extension to cases in which gravity also comes into play. In a similar vein, I found not quite convincing and a bit artificial the suggested contrast between internal models and optimal control. In fact, internal models can be considered in the framework of optimal control, since they enable to predict the consequences of motor commands (forward models) and to form different optimal solutions of ill-posed inverse problems. What the authors consider as kinematics compensations (i.e. the formation of minimum jerk or minimum torque change trajectories) should not be described as alternative to optimization since they are also forms of optimization.

3)The alternative between "compensation" and "optimization" is not clear as compensation is often described (or describable) as optimization.

4)"SR dependence on movement direction (i.e., on gravity torque) is a unique feature of effort related optimization." Is there a proof for this? Could some similar dependence result from different coordinate representation (as for example min-jerk expressed in angular coordinates)?

[Editors' note: further revisions were requested prior to acceptance, as described below.]

Thank you for resubmitting your work entitled "Direction-dependent arm kinematics reveal optimal integration of gravity cues" for further consideration at *eLife*. Your revised article has been favorably evaluated by Eve Marder (Senior editor), a Reviewing editor, and three reviewers. We are sorry for the delay in returning this to you, but your manuscript has engendered a lot of discussion among the reviewers (meaning of course that they care about the work!).

The manuscript has been improved but there are some remaining issues that need to be addressed. I am giving you the reviews in full, so you can see the issues that are being raised by the reviewers:

Reviewer #1:

The authors have significantly improved the manuscript, and I am satisfied with the revision.

Reviewer #2:

The authors have adequately addressed several concerns. The manuscript now more completely specifies the present smooth effort model, which makes the whole argument easier to understand. The additional modeling results, including demonstration of their torque change model generating varying symmetry ratios under varying duration and load, is heartening.

At this point there are two competing considerations for the manuscript as a whole.

1) The connection between theory and experiment in Figure 1 and Figure 2 do suggest that "effort" likely plays a role in movement asymmetry. The gradual reduction of this asymmetry in parabolic flight does suggest that different environmental conditions lead to alteration of neural control. On the whole, then, this does forward the field and provide interesting insight.

2) Several concerns remain. Even with the additional exploration of their own implementation of the torque change model, the absolute flatness of model results in Figure 1 seems unrealistic. The authors could make a much more compelling case: they could replot the y-axis data (experimental symmetric ratio) versus calculated torque change. If this experimental result is just as flat at 1F, that would be convincing. If not, then the authors need to reconsider the uniqueness of their model at explaining the result.

The reconsideration of the minimum variance model is far more concerning. [Disp-formula equ10] removes any angular dependence from gravity. This step seems to prevent any meaningful outcome from their exploration of how angle influences control.

One comment from another reviewer has resonated with me. There is an alternate interpretation to early movements in parabolic flight, in which asymmetries mimic those on the ground. The reviewer rightly points out that actually experienced dynamics are very different, so the replication of asymmetry in kinematics is very odd. At the very least, this is evidence that actually-experienced gravity (or lack thereof) does not meaningfully influence control, and that the whole interpretation here is questionable. (It is a very curious result, at least).

Taking the clear with the unclear, I do think that the overall results forward the field, although I do have ongoing concerns with the models and math, as presently implemented.

Reviewer #3:

The authors have addressed several of the concerns in the revised manuscript, resulting in an improvement of the final product. The main element of novelty and interest of this work remains the demonstration – through the 0 g experiment – that the asymmetry in the kinematics of vertical reaching reflect a process of adaptation to the gravitational field. The effect is small, but I agree with the authors that this relatively small and specific effect is likely to reflect a general evolutionary (but also ontogenetic) process that rewards the ability of an organism to take advantage of the characteristics of its environment.

My major concerns remain with the attempt to overgeneralize this findings and suggest that it disproves what the authors call "compensation hypothesis" in a broader sense. This sounds a bit like a political argument, where one takes the sides of a dynamical optimization against geometrical optimization. The fact is that the literature has shown evidence of both in different contexts. A case in point can be found in a recent study by Farshchansadegh and colleagues (PLoS Computational Biology 2016) who found evidence for both dynamic ad kinematic optimization under the same mechanical conditions, depending upon the agreement between visual and haptic feedback. That said, I am normally convinced that the essential value of a manuscript should be based on the results, not on how the authors want to organize the discussion. Broad speculations about the findings are a matter of choice and intellectual freedom of the authors. However, one should avoid statements that are inaccurate, such as the suggestion that "the compensation hypothesis has critically lacked quantitative support." Perhaps this statement could only be justified by the fact that most researchers proposing that there are geometrical constraints in shaping the planning of multi articular reaching movements have not referred to this ide as "the compensation hypothesis".

---

## [Author Response]

[…]

Reviewer #1:

[…]

This study continues the series of nice papers published by the authors on the effects of gravitoinertial constraints on sensorimotor processes. It addresses an important issue related to the integration of gravity during motor actions and provides significant advance over previous studies. However, the paper could be easily improved with some changes.

The problem is that other published papers, some of them referenced in the manuscript, provided convincing evidence in favor of the compensation hypothesis. Unfortunately, the authors do not provide explanations as to why the earlier evidence for the compensation hypothesis is less convincing than theirs. They kind of "kicked the ball into touch" by simply stating that "advent of the optimal control theory has cast strong doubts on this more traditional view of motor control". This was done perhaps on a voluntary basis because critics, even those that lead to progress, are not always welcome. However, because the goal of the study was "to explicitly distinguish between the compensation and optimization hypotheses", the discussion should be more explicit about problems regarding the compensation hypothesis. Despite the very nice results, without such a discussion, the impact of the present paper will be dampened.

We have now modified the Discussion to address this comment. We state that “Although very influential, such compensation hypothesis has lacked quantitative support. In contrast, other studies that quantified velocity profiles revealed that the temporal organization of arm kinematics shows a small, yet consistent, dependence on movement direction, speed and load (Gaveau et al., 2014, Gaveau and Papaxanthis, 2011, Papaxanthis et al., 1998b). Furthermore, findings inconsistent with the compensation hypothesis (e.g., negative phasic EMG activity during vertical arm movements; Flanders et al., 1996) have been ignored” (Discussion paragraph five). Previous studies that attempted to quantitatively validate the compensation hypothesis reported complex results that were not evaluated quantitatively with alternative hypotheses. Thus, even results inconsistent with the compensation hypothesis have been ignored. The recent application of optimal control theory to biological movement have allowed better quantification and interpretation of complex data sets, including the notion that motor adaptation corresponds to a re-optimization process giving rise to altered trajectories.

The writing style would benefit from smoothing. Currently, there are too many long sentences, many of them are interrupted by long lists of references that sometimes interfere with understanding.

We have tried to make the writing smoother throughout the paper.

Reviewer #2:

The authors provide interesting evidence of gravitational influence on human single-joint movements. The directional dependence on the ground, and acquired loss of dependence in repeated parabolic flight, is substantive and novel. This work has implication for how the brain can ordinarily (on the ground) and flexibly (via parabolic flight) be influenced by gravity.

The particular analytical, theoretical, and hypotheses offered, however, have several under-developed and contradictory elements that cause great concern. The chief concern, with regard to the overall hypothesis, regards the sureness that these results confirm a "Minimum Smooth Effort" model. The authors choose to cite papers that develop the model in full, rather than doing so here; that is a great disadvantage, as the reader cannot even consider the actual hypothesis suggested.

We thank the reviewer for pointing this out. In the revised manuscript, we have developed the “Simulations” section more carefully such that the reader can easily follow the logic that is provided with all required materials to reproduce the simulations. The length of the “Simulations” section was increased. More specific remarks on simulations are addressed in the following comments.

What is suggested throughout is that "effort" is dominated by forces (here, gravity) (e.g. "The behavioral signature of the optimal integration of gravity hypothesis…") The equations offered, however, show that jerk, a purely kinematic measure, is included in the authors' model. The sensitivity analysis, shown only in the discussion, is very incomplete, as it explores the calculated cost only on one side of the chosen coefficient. My best conclusion is that the jerk inclusion is crucial, and even the most generous reading of the work concludes with a synthesis, that both kinematics and dynamics are important in the present behavior.

The reviewer’s intuition is not true. Figure 1—figure supplement 2 shows that the asymmetry is gradually reduced as the relative weight of the jerk over the effort is increased. This demonstration is sufficient for the following reason: When resolving an optimal control problem, the cost function is defined up to a positive scalar factor. More precisely, if one minimizes some functionf(x)=b∗g(x)+c∗h(x)

the solution x will be the same as if one minimizesf(x)/d=b/d∗g(x)+c/d∗h(x)

for any d>0. Therefore, we can normalize our cost function by setting b=d, thereby assuming a unit coefficient in front of the absolute work component in our case. Varying our unique weighting factor is sufficient to explore the two extreme cases, namely jerk alone and absolute work alone. When the weighting factor is zero (or very small in practice), we minimize the absolute work alone. When the weighting factor is infinite (or very large in practice), we minimize the jerk alone. Hence, our sensitivity analysis is sufficient to demonstrate that the capacity of the smooth-effort model in predicting the sigmoidal effect of direction on SR emerges from the effort part of the cost only. We now clearly explain this point in the subsection “Simulations”.

*Several other elements of the analysis, modeling, and interpretation are of concern.*

– The presented smooth effort model includes separate terms for torques generated by agonists and antagonists. The authors have no way to measure these; they measure only kinematics; undoubtedly on the ground, and absolutely in flight, there will be co-contraction. The central variables in their model, therefore, are unmeasurable.

Because we measure kinematics only, we indeed have no measure of agonist versus antagonist muscles participation to the motion of the arm; we can only infer the net torque. However, since agonist and antagonist torques are opposed and angular velocity keeps a constant sign (e.g. non-negative for upward motion), minimizing the absolute work of the muscle torque whilst separating agonist and antagonist torque or not (working on the net torque directly) is equivalent: this does not affect the prediction of the Minimum Smooth-Effort model. We have already shown this in a previous paper (Berret et al. 2008, PLoS Comp Biol). This is a logical result since, in both cases, minimizing the absolute work of forces will result in a control policy that produces as little co-contraction as possible. Separating agonist and antagonist torques would be interesting to test the effect of introducing some muscular dynamics (as we additionally did for the Minimum Variance Model, please see below), however, to allow meaningful interpretation of our results and reduce the number of unknown/tunable parameters, we used the simplest minimum smooth effort model without any muscular contraction dynamics. In doing so, the predicted directional asymmetries can surely be attributed to the rules of physics, i.e. gravity effects, and not to the specific muscular dynamics that we would implement. Accordingly, in the new version of the manuscript, we rewrote the equations of the model with the net torque only (please see [Disp-formula equ4 equ5 equ6]). We thank the reviewer for this constructive remark as it allows a simpler interpretation of our findings.

– The absolute flat lines calculated from minimum torque change and minimum variance hypotheses (Figure 1) is highly questionable. Similar work (indeed the origin of these theories) suggest that load should play an important role. Much more detail on the calculation of these cost functions is needed to justify this very counter-intuitive result. The minimum variance model, in particular, needs a level of neural activation (drive) that seems complete absent from the present consideration.

The minimum Variance model is based on the principle that some signal-dependent noise – multiplicative noise that scales with the muscle contraction level (no constant noise because duration is defined a-priori here) – corrupts muscle activation patterns and thus end-point accuracy. In our initial version of the manuscript, we did mention that our simulations incorporate signal dependent noise, however, our description of the Minimum Variance model was very brief and did not even define the control variable. Here-below we provide the necessary information to prove the effectiveness of our minimum variance simulations.

We took advantage of the study from Tanaka and collaborators (2004, Neural Computation, paper communicated by Daniel Wolpert) where the authors have compared the predictions of the minimum Variance and the Minimum Torque change models for mono-articular arm movements performed under various conditions of movement duration and external force fields. First, in agreement with our results, Tanaka and colleagues found that the minimum Torque Change model predicts velocity profiles that are always symmetrical (i.e. Symmetry Ratio = 0.5). Second, the minimum Variance was found to predict symmetry ratios that increased with movement duration and that decreased when increasing the system degree of stability (when movements are performed in hybrid viscous and elastic force fields).

Figure 4 here below presents the results of our own equivalent simulations. Proving the validity of our minimum Variance formulation, it can be observed that our simulations reproduce both previously described effects (see Tanaka et al. 2004, Figure 2 and Figure 3).

Given the effect of the hybrid viscous and elastic force fields on SR, we understand the reviewer’s surprise with regard to our result of invariant SR predicted by the minimum Variance when movement direction changes in the gravity field. However, the gravitational field is very different from the hybrid viscous and elastic force field of Tanaka et al. (2004) in the sense that the system’s degree of stability does not change with movement direction. The signal dependent noise (i.e. variance), accumulating with the muscular contractions responsible for movement acceleration and deceleration, actually reaches a constant final level for all movement directions in the gravity field (for a given duration and system’s stability). As a result, velocity profiles, quantified by SR, are similar in all directions. This is a coherent result since the minimum variance penalizes large muscular torques that would be necessary to reproduce our experimental results; i.e. a short SR when moving upward (requiring strong acceleration against gravity and therefore high noise) and a long SR when moving downward (requiring strong deceleration against gravity and therefore high noise).

In the revised manuscript, we have provided further details in the Methods related to the minimum Variance simulations. The reader should now appreciate that our formulation of the minimum Variance model is similar to previous studies (Subsections “Simulations”). We have also added simulations showing that including some simple muscle dynamics (making the muscle torque a low-pass filtered version of the control signal) into the optimal control problem (as performed by Harris and Wolpert, 1999) did not change the conclusions made from data presented in the main text (see subsection “Simulations” and Figure 1—figure supplement 3). Note that Tanaka et al. (2004) also reported that including muscle dynamics into their simulations did not change the conclusions of their analysis.

Author response image 1.Additional simulations testing the validity of our minimum Variance simulations by comparison to previously published results from Tanaka et al. 2004.(**A**) Normalized velocity profiles obtained for various movement durations. (**B**) Symmetry Ratio was shown to increase with movement duration (**C**) Normalized velocity profiles obtained for various degrees of system’s stability. The right-most curve corresponds to the case of no external force. The other four curves, from right to left, are for critical damping system with increased stability via external viscous and elastic forces (see Tanaka et al. 2004). (**D**) Symmetry Ratio was shown to decrease when the force field level increased.**DOI:**
http://dx.doi.org/10.7554/eLife.16394.011

– Passengers that are strapped in the flight clearly have cues that gravity is very strong, absent, and strong again. The notion that strapping will lead to no information before movement is vexing. Similarly, the notion that hypergravity for the not-strapped-down has no learning effect is unusual.

Our rationale was that preventing subjects from performing any voluntary movements (while they were strapped on the floor and between 0G phases for the not-strapped-down subject) would allow reducing sensory-motor adaptation by preventing motor interaction with the various unwanted gravito-inertial force fields. We observed that subjects similarly adapted across parabolas independently of whether they started the experiment at the beginning of the flight or waited strapped on the floor (as stipulated in the manuscript subsection “Experiment 2”). These results therefore demonstrate the effectiveness of the parabolic flight experimental paradigm to answer our question. In the revised manuscript, we replaced “avoid” by “reduce” in the same section. We also added a supplemental figure (Figure 3—figure supplement 1) allowing the reader to compare results of the 1st and 2nd subject of each flight. Furthermore, a Kruskal-Wallis ANOVA on ranks did not reveal any difference between the two groups (subjects passing 1st or 2nd).

– The authors provide scant information on how they calculated just the biomechanics of the task. Meaningful quantitative consideration of kinematics versus dynamics requires more information here.

We updated Figure 1—figure supplement 1 to provide full details on how Gravity torque ([Disp-formula equ1]) could be computed from arm kinematics. We refer to it in the manuscript: “The geometrical illustration of the mechanical system along with details on how to derive [Disp-formula equ1] are displayed in Figure 1—figure supplement 1“.

On the whole, the authors have shown that in a vertical plane, target direction, on the ground, plays a role in human trajectory control; that in early flights, an asymmetry in velocity profiles exist; and in later flights, that asymmetry disappears. All these point to a simple conclusion: the dynamics of gravity can be used by the brain to influence trajectory. All further conclusions made by the manuscript have under-developed evidence, and therefore provide very unclear impact on the field of motor control and neural computation.

We hope the reviewer finds that the revised paper addresses his concerns and supports our conclusion that the brain optimizes the mechanical effects of gravity to minimize movement effort. We believe this is an important finding which may have strong implications in fields as diverse as neurorehabilitation, movement perception or motor control modularity, which for the most part assume the compensation principle (Prange et al. 2009; Prange et al. 2012; Cook et al., 2013; Russo et al. 2012).

Reviewer #3:

[…]

Substantive concerns

1) Small effect in a limited domain of motor behavior. The effect considered is a relatively subtle asymmetry in the speed profile that is consistent with exploiting the gravitational torque in the upward and downward phase of movement. While the data are consistent with the postulated cost function, the argument about the evolutionary advantage does not seem to match the small size of the described effect. Perhaps the effect on higher dimensional motions would be more substantial.

Because human muscles present temporal constraints regarding their force production capacity (Winters JM and Stark L, IEEE Trans Biomed Eng, 1985; Zajac FE, Crit Rev Biomed Eng, 1989), SR cannot be expected to change too drastically. Although the maximal difference between upward and downward SR is small, all subjects showed a similar trend (Figure 2) and the statistical significance was high.

Although beyond the scope of this paper, for an example of more substantial effects in higher dimensional motion, please see preliminary results presented in Figure 5 (explained below).

In the present study, we focused on single degree of freedom arm movements (thus on the shape of speed profile) to specifically isolate gravity effects. Previous studies, however, uncovered similar effects of movement direction (upward/downward) both on spatial and temporal aspects of multi-degree of freedom movements at various speeds and amplitudes, for arm and for whole body (sit to stand versus stand to sit) movements (Papaxanthis et al., 2005, Yamamoto and Kushiro, 2014, Papaxanthis et al., 2003, Papaxanthis et al., 1998c). Importantly, the minimization of movement effort in the gravity field can explain these differences (Berret et al. 2008; Gaveau et al. 2011, 2014).

Because results in the current study were highly significant and because previous studies have uncovered similar effects in various domains of motor behavior, we believe it is reasonable to propose that the versatility of motor planning in the gravity field results from an evolutionary and/or developmental advantage for motor economy. This argument is developed in the Discussion section. We have now replaced “argue” by “propose” and added “and/or developmental”.

2) The model does not refute the minimum jerk optimization but it combines it with the optimization of work against gravity. The smoothness model was proposed for planar horizontal movements that are unaffected by gravity. In this respect the model of Gaveau and colleagues does not seem an "alternative" to the minimum jerk model but an extension to cases in which gravity also comes into play.

We now make this point at the end of the Discussion. It is important to note that, for horizontal movements, our model makes predictions that are close but not exactly equal to those of the minimum Jerk model. For example, because the effort part of our model is torque dependent, the prediction of the Smooth-Effort model in the horizontal plane changes with the anthropometric characteristics of each subject (itself changing the inertial torque level). In general, the prediction of the Smooth-Effort model will change with the torque requirement of the movement whilst the minimum Jerk prediction will not. Therefore, if interaction torques (for multi-degree of freedom arm movements) or additional external torques (produced by a robotic manipulandum for example) are experienced in the horizontal plane, the minimum Smooth-Effort and the minimum Jerk would predict very different solutions. This is now mentioned in the Methods that develop the models (see subsection “Simulations”).

Actually, we have collected preliminary data to test this hypothesis: subjects perform 60 multi-articular reaching arm movements (all joints free of constraint; movements performed between classical sets of two targets; i.e. starting and final) in the horizontal plane while handling a robotic manipulandum applying external assistive, resistive or no force on the subject’s arm. Figure 5 shows the stabilized performance (average of last 30 trials, +SD) for the first three subjects who performed this experiment. It is striking that the temporal organization of the reaching movement is adapted such that symmetry ratios are smaller for resistive (same as upward in the current paper) and larger for assistive (same as downward in the current paper) compared to the no force condition (similar to horizontal in the current paper).

Therefore, although we agree that the present version of the minimum Smooth-Effort model does not refute the minimum Jerk optimization, as it combines the minimum Jerk with an Effort cost, we believe that our model does not only extend to cases in which gravity comes into play but actually represents a generalization that accounts for how task dynamics shapes movement kinematics.

Author response image 2.Symmetry ratios (average of 30 trials, +SD) obtained for three subjects performing fast arm reaching movements in the horizontal plane under three conditions of external force.Assistive: the manipulandum pushes the arm toward the final target. No force: the manipulandum does not exert any force. Resistive: the manipulandum pushes the arm toward the starting target. Movement amplitude was 30 cm and duration ≅ 300ms.**DOI:**
http://dx.doi.org/10.7554/eLife.16394.012

In a similar vein, I found not quite convincing and a bit artificial the suggested contrast between internal models and optimal control. In fact, internal models can be considered in the framework of optimal control, since they enable to predict the consequences of motor commands (forward models) and to form different optimal solutions of ill-posed inverse problems. What the authors consider as kinematics compensations (i.e. the formation of minimum jerk or minimum torque change trajectories) should not be described as alternative to optimization since they are also forms of optimization.

It was not our intention to contrast internal models with optimal control. We completely agree with the reviewer about the necessary consideration of internal models in the optimal control framework. We have now modified the Introduction and introduce the two opposed hypothesis (compensation vs effort optimization), both within the optimal control framework. In both cases we explicitly state that the internal model of gravity will benefit one strategy or the other by allowing anticipation of gravity effects.

3)The alternative between "compensation" and "optimization" is not clear as compensation is often described (or describable) as optimization.

The text has been modified to avoid confusion. We use the optimal control framework to disentangle between two alternative hypotheses on how the brain uses the gravity internal model when planning arm movements – i.e. we use various minimization criteria to simulate control policies that compensate or optimize the mechanical effects of gravity. A purely kinematic optimization (such as the minimum Jerk) must take gravity effects into account in order to compensate them and therefore produce the desired trajectory (the smoothest one). On the other hand, an optimization criterion that considers movement dynamics will take advantage of gravity effects in order to produce trajectories that satisfy the desired minimization: endpoint variance, muscular effort or else. Therefore, in this study, we contrast kinematic and dynamic costs to disambiguate the compensation versus optimization hypothesis of gravity effects. Then we compare the predictions of various dynamic costs to emphasize the importance of effort in motor planning (Effort versus Variance or Torque Change).

We realize that contrasting an optimal control model (the minimum jerk) with the “optimization” hypothesis of gravity effects was confusing. For the sake of clarity in the new version of the manuscript we replaced the name “optimization hypothesis” by the “effort optimization hypothesis”. Additionally, in the Introduction and Results section we explicitly state that we use optimal control models to illustrate both gravity effect compensation and gravity effect optimization (i.e. the effort optimization hypothesis).

4)"SR dependence on movement direction (i.e., on gravity torque) is a unique feature of effort related optimization." Is there a proof for this? Could some similar dependence result from different coordinate representation (as for example min-jerk expressed in angular coordinates)?

Amongst the various models simulated to test whether they can reproduce kinematic asymmetries in the vertical plane, only models that minimized an effort-related cost succeeded (Berret et al. 2008a,b; Crevecoeur et al. 2009; Gaveau et al. 2014). Furthermore, in previous studies (Berret et al. 2008a,b) an inverse optimal control approach mathematically demonstrated that the presence of an effort-related criterion in the cost function is a necessary and sufficient condition for the production of a brief transient muscle inactivation near the peak of velocity which induces the observed kinematic asymmetries with respect to movement direction. We clearly stipulate this in the revised manuscript. In addition, direct optimal control simulations confirmed that other costs such as the integrals of squared torque change, torque, and jerk (Cartesian or angular) cannot consistently reproduce the robust empirical up/down asymmetries. Regarding the use of different coordinate representations, Cartesian and angular are identical for a single degree of freedom movement. This last point was added to the manuscript subsection “Simulations”.

*[Editors' note: further revisions were requested prior to acceptance, as described below.]*

[…]

Reviewer #2:

The authors have adequately addressed several concerns. The manuscript now more completely specifies the present smooth effort model, which makes the whole argument easier to understand. The additional modeling results, including demonstration of their torque change model generating varying symmetry ratios under varying duration and load, is heartening.

At this point there are two competing considerations for the manuscript as a whole.

1) The connection between theory and experiment in Figure 1 and Figure 2 do suggest that "effort" likely plays a role in movement asymmetry. The gradual reduction of this asymmetry in parabolic flight does suggest that different environmental conditions lead to alteration of neural control. On the whole, then, this does forward the field and provide interesting insight.

We thank the reviewer for her/his positive comments.

2) Several concerns remain. Even with the additional exploration of their own implementation of the torque change model, the absolute flatness of model results in Figure 1 seems unrealistic. The authors could make a much more compelling case: they could replot the y-axis data (experimental symmetric ratio) versus calculated torque change. If this experimental result is just as flat at 1F, that would be convincing. If not, then the authors need to reconsider the uniqueness of their model at explaining the result.

The reviewer is still not convinced by our simulations of the minimum Torque Change model. We would like to underline that other authors (Tanaka et al. 2004) also demonstrated that optimizing the torque change predicts constant velocity profiles (i.e., symmetry ratios) for various movement conditions (changing orientation, speed or else…). To clarify this result, we provide, at the end of the response letter, a formal demonstration that the Torque-change model does not predict varying symmetry ratios.

During our investigations, we did question ourselves about why other models do not account for changes in symmetry ratio as a function of movement direction. What the reviewer is asking for, plotting the experimental symmetric ratio versus the calculated torque change derived from the experimental velocity profiles (if we understood correctly), would not help. Correlating variables that do not result from a model prediction is inappropriate to test whether a theoretical model (here the Torque Change model) can explain our experimental data or not; i.e., to reconsider the uniqueness of our model at explaining the experimental results. This would only show that the torque change (computed from experimental data) correlates with arm kinematics. This would not mean that the torque change variations are optimal and thus correspond to a planning principle, which is the aim of our study. Our formal demonstration below, along with the simulations presented in the manuscript and previous results published by Tanaka et al. (2004) prove that the torque-change predictions are “flat”.

The reconsideration of the minimum variance model is far more concerning. [Disp-formula equ10] removes any angular dependence from gravity. This step seems to prevent any meaningful outcome from their exploration of how angle influences control.

[Disp-formula equ10] does not totally remove angular dependence from gravity. Indeed, there are two different angular dependences regarding gravity: one from the movement plane orientation (𝛾 in our manuscript) and one from the time-varying movement amplitude (𝜃 in our manuscript). The work of gravity torque (as presented in Figure 1) encompasses both of these angular dependences.

[Disp-formula equ10] removes angular dependence from movement amplitude only; i.e., the modulation of the projected gravity torque (PGT) that is observed between 𝜃 = 0° and 𝜃 = 45° from each plot in Figure 1—figure supplement 1 (panel B). However, the cosine function varies from 1.0 to 0.7 on that interval and it is not this relatively small variation that is expected to account for the values of the symmetry ratio. The influence of movement plane orientation is actually much more important. This is illustrated by Figure 1—figure supplement 1 (panel B), where it can be observed that the angular dependence of PGT on movement amplitude mostly concerns movements performed in vertical movement planes (𝛾 > 45° 𝑎𝑛𝑑 𝛾 < −45°). For the other movement planes (𝛾 > 45° 𝑎𝑛𝑑 𝛾 < −45°), there is almost no modulation of PGT through movement amplitude. The mean (over the 17 targets) modulation of PGT, which we neglect with [Disp-formula equ10], is equal to 11.88% (1.42N.m on average) of the PGT experienced on the starting target. For comparison, reversing movement orientation from upwards (𝛾 = 90°) to downwards (𝛾 = −90°) produces a 200% PGT modulation. Importantly, [Disp-formula equ10] does not neglect this plane orientation angular dependence from gravity. Figure 6 illustrates how small effects the approximation given in [Disp-formula equ10] has on the Work of Gravity Torque, the experimentally manipulated parameter in our single-joint movement paradigm. It is important to note that, in our experiment, the main effect of gravity is due to movement plane orientation (given by angle 𝛾), not to the time-varying movement amplitude 𝜃.

To further illustrate this effect on optimal control simulation, we also present the results of two sets of simulations performed with the Smooth-Effort model in the case where we neglect the angular effect of movement amplitude (as in [Disp-formula equ10] for the Variance model) and in the case where we do not neglect the angular effect of movement amplitude (Figure 6). It can be appreciated that this approximation does not qualitatively change the results; i.e., the symmetry ratio still linearly correlates with the work of gravity torque. In conclusion, the approximation of [Disp-formula equ10] does not remove the main angular dependence from gravity, i.e., according to the plane of motion, and the differences between the results obtained from the Smooth-Effort and the Variance models cannot be attributed to the approximation made in [Disp-formula equ10].

Author response image 3.(**A**) Effect of [Disp-formula equ10] approximation on the work of gravity torque. Empty black circles depict the work of gravity torque (as a function of movement direction; i.e., plane angle 𝛾) computed with the same approximation as [Disp-formula equ10]; i.e., when assuming that gravity torque is constant throughout movement amplitude. Colored circles depict the work of gravity torque without any approximation (same data as in Figure 1). (**B**) Effect of [Disp-formula equ10] approximation on *Smooth-Effort* simulated symmetry ratios as a function of WGT. Black dots depict simulated data with the same approximation as [Disp-formula equ10]; i.e., when assuming that gravity torque is constant throughout movement amplitude. Red dots depict simulated data without approximation (same data as in Figure 1). Each data point represents one subject moving in one direction (n=255 in each plot).**DOI:**
http://dx.doi.org/10.7554/eLife.16394.013

One comment from another reviewer has resonated with me. There is an alternate interpretation to early movements in parabolic flight, in which asymmetries mimic those on the ground. The reviewer rightly points out that actually experienced dynamics are very different, so the replication of asymmetry in kinematics is very odd. At the very least, this is evidence that actually-experienced gravity (or lack thereof) does not meaningfully influence control, and that the whole interpretation here is questionable. (It is a very curious result, at least).

We agree with the reviewer that the in-flight early persistence of directional asymmetries is a non-trivial result. However, to understand how the brain integrates gravity torque, one must consider the whole adaptation pattern, i.e., the initial values in 0g along with the following ones, not only the initial or final ones. Existing literature on motor adaption to microgravity or, more generally, to a new dynamical environment, allows better interpreting this result.

In the light of previous microgravity studies (Crevecoeur F. McIntyre J., ThonnardJ.-L and Lefèvre P., 2010 and 2014), such behavior (the persistence of 1g behavior in early 0g followed by progressive adaptation toward new values) could be interpreted as a stabilization of movement performance under internal model uncertainty. For example, in their 2010 paper, the above-mentioned authors observed that the grip force required to holding an object in the hand (pinched between thumb and index fingers) was kept similar to 1g values at the beginning of 0g exposition. Then, grip force was observed to slowly decrease toward new optimal values across parabolas. Theoretical simulations revealed that this adaptation dynamics reflects the stabilization of movement performance when the internal model of gravity is changing (uncertain). More generally, this stabilization concords with the first of a dual adaptation process previously proposed by Izawa J., Rane T., Donchin O. and Shadmehr R., 2008: “on the one hand, adaptation produces a more accurate estimate of the sensory consequences of the motor commands (i.e., learn an accurate forward model), and on the other hand, our brain searches for a better movement plan so to minimize an implicit motor cost and maximize rewards (i.e., find an optimum controller)”.

Reviewer #3:

*[…]*

My major concerns remain with the attempt to overgeneralize this findings and suggest that it disproves what the authors call "compensation hypothesis" in a broader sense. This sounds a bit like a political argument, where one takes the sides of a dynamical optimization against geometrical optimization. The fact is that the literature has shown evidence of both in different contexts. A case in point can be found in a recent study by Farshchansadegh and colleagues (PLoS Computational Biology 2016) who found evidence for both dynamic ad kinematic optimization under the same mechanical conditions, depending upon the agreement between visual and haptic feedback. That said, I am normally convinced that the essential value of a manuscript should be based on the results, not on how the authors want to organize the discussion. Broad speculations about the findings are a matter of choice and intellectual freedom of the authors. However, one should avoid statements that are inaccurate, such as the suggestion that "the compensation hypothesis has critically lacked quantitative support." Perhaps this statement could only be justified by the fact that most researchers proposing that there are geometrical constraints in shaping the planning of multi articular reaching movements have not referred to this ide as "the compensation hypothesis".

We understand the reviewer’s remark and we thank her/him for the respect of our intellectual freedom. To avoid inaccurate statements, we modified the sentence. We now say: “such a compensation hypothesis has been challenged by results of studies that quantified velocity profiles and revealed that the temporal organization of arm kinematics shows a small, yet consistent, dependence on movement direction, speed and load”.

We also thank the reviewer for pointing out the very interesting and newly published study from Farshchiansadegh and colleagues (2016). We believe this study does not support the existence of a kinematic optimization per se but shows that different sensory perception of the environment (visual vs proprioceptive) can lead to energy optimization that is not always dissociable from kinematic optimization. We referenced this study as well as another newly published one (Shadmehr et al. 2016) on the importance of effort in tailoring motor behaviors.